# Parental Perceptions and Barriers towards Childhood COVID-19 Vaccination in Saudi Arabia: A Cross-Sectional Analysis

**DOI:** 10.3390/vaccines10122093

**Published:** 2022-12-07

**Authors:** Yusra Habib Khan, Tauqeer Hussain Mallhi, Muhammad Salman, Nida Tanveer, Muhammad Hammad Butt, Zia Ul Mustafa, Raja Ahsan Aftab, Abdullah Salah Alanazi

**Affiliations:** 1Department of Clinical Pharmacy, College of Pharmacy, Jouf University, Sakaka 72388, Saudi Arabia; 2Health Sciences Research Unit, Jouf University, Sakaka 72388, Saudi Arabia; 3Institute of Pharmacy, Faculty of Pharmaceutical and Allied Health Sciences, Lahore College for Women University, Lahore 54000, Pakistan; 4Institute of Molecular Cardiology, University of Louisville, Louisville, KY 40202, USA; 5Department of Medicinal Chemistry, Faculty of Pharmacy, Uppsala University, 75123 Uppsala, Sweden; 6Discipline of Clinical Pharmacy, School of Pharmaceutical Sciences, Universiti Sains Malaysia, Gelugor 11800, Malaysia; 7Department of Pharmacy Services, District Headquarter (DHQ) Hospital, Pakpattan 57400, Pakistan; 8Department of Clinical Pharmacy and Pharmacy Practice, Faculty of Pharmacy, Universiti Malaya, Kuala Lumpur 50603, Malaysia

**Keywords:** COVID-19, SARS-CoV, pandemic, vaccines, hesitancy, parents, pediatrics, children, vaccination, childhood

## Abstract

Introduction: The vaccination of children against Coronavirus disease (COVID-19) is a prime area of focus around the globe and is considered a pivotal challenge during the ongoing pandemic. This study aimed to assess parents′ intentions to vaccinate their children and the barriers related to pediatric COVID-19 vaccination. Methodology: An online web-based survey was conducted to recruit parents with at least one child under the age of 12 years from Saudi Arabia’s Al-Jouf region. The parental intentions to vaccinate children were assessed via six items, while barriers against vaccination were assessed through seven items in validated study instrument. A 5-point Likert scale was used to record the responses of parents regarding both their intentions and barriers. Results: In total, 444 parents (28.41 ± 7.4 years, 65% females) participated in this study. Almost 90% of parents were vaccinated against COVID-19 but only 42% of parents intended to vaccinate their children. The mean intention score was 2.9 ± 1.36. More than one-third of study participants had no plan to vaccinate their children against COVID-19. The majority of the respondents agreed to vaccinate their children if vaccination was made compulsory by the government (relative index: 0.76, 73%). Out of seven potential barriers analyzed, concerns over vaccine safety and side effects were ranked highest (RII: 0.754), reported by 290 (65%) participants. In multivariate logistic regression, significant predictors of parental intention to vaccinate children were the increased education level of the parents (secondary education: OR = 3.617, *p* = 0.010; tertiary education: OR = 2.775, *p* = 0.042), COVID-19 vaccination status (vaccinated: OR = 7.062, *p* = 0.003), mother’s involvement in decisions regarding the child’s healthcare (mother: OR 4.353, *p* < 0.001; both father and mother: OR 3.195, *p* < 0.001) and parents’ trust in the vaccine’s safety (OR = 2.483, *p* = 0.022). Conclusions: This study underscored the low intention among parents to vaccinate their children against COVID-19. Vaccination intention was found to be associated with education, parents’ vaccination status, the mother’s involvement in healthcare decisions, and parents’ trust in the vaccine’s safety. On the other hand, parents’ concerns over the safety and efficacy of the COVID-19 vaccine were widely reported as barriers to childhood vaccination. The health authorities should focus on addressing parental concerns about vaccines to improve their COVID-19 vaccination coverage.

## 1. Introduction

The management of the coronavirus disease (COVID-19) pandemic has shifted from precautionary measures such as social distancing, masks, curfews, financial penalties and lockdowns to vaccination of the adult population. The COVID-19 vaccination has been regarded as a largest vaccination program against an infectious disease. According to the World Health Organization (WHO), an estimated population of 5 billion had been fully vaccinated and about 5.4 billion had been partially vaccinated by 17 November 2022 [1]. The ongoing efforts towards managing COVID-19 urge the initiation of vaccination among children and adolescents to improve the coverage. Canada was the first country to approve the COVID-19 vaccine for adolescents in May 2021. Subsequently, the childhood vaccination was approved in United States of America (USA) and the United Kingdom (UK) [2]. Currently, almost all the countries are struggling to vaccinate majority of their population including children. Although the COVID-19 infection is mild and mostly asymptomatic in children; however, in some cases, it might lead to multi-system inflammatory syndrome (MIS) [3]. Moreover, new variants of the virus, along with failure to achieve the targets of herd immunity in various regions, also necessitate vaccination against COVID-19 among children [4,5]. Owing to the different immunological responses, special needs and features in children, the safety and efficacy of adult COVID-19 vaccines cannot be assumed in children [6]. Research studies have highlighted the safe outcomes of COVID-19 vaccines among those aged 18 years and above, with only mild side effects. Most of these side effects were transient in nature. The frequency of these side effects has been found to range from 28% to 52% for fatigue, up to 47% for injection site pain, 31% to 37% for fever, up to 23% for muscular pain and up to 32% for headache [7,8,9]. Contrary to the majority of the studies reporting the positive outcomes of the COVID-19 vaccine in children, some studies have also reported cardiac inflammation [10,11]. Since the beginning of the pandemic, the global prevalence of COVID-19 among children has accounted for 1–5% of all reported cases, while 8% of all cases were reported among children in Saudi Arabia [12,13,14]. A successful vaccination campaign and herd immunity cannot be achieved without vaccination of children, as they account for a substantial proportion of the total population.

Beyond the multifaceted logistics of the vaccine’s manufacturing, testing, distribution and quality assurance, the acceptance of the COVID-19 vaccine is a major global challenge. Vaccine hesitancy is not a new phenomenon. The World Health Organization (WHO) has listed it as one of the major global health threats [15]. The expanded program of immunization (EPI) developed by the WHO was first implemented in 1979 in Saudi Arabia [16]. Various studies have reported parents’ hesitancy (approximately 25%) toward routine vaccination in Kingdom of Saudi Arabia (KSA) [17,18,19]. A low routine pediatric vaccination rate has been observed in the Al-Jouf region as compared with other developed cities in KSA [20]. Delays in routine pediatric immunization and annual influenza vaccination are a long-standing public health concern that might also lead to hesitancy toward the childhood COVID-19 vaccine.

As of November 18, 2022, Saudi Arabia has administered about 69 million doses of the COVID-19 vaccines [21]. According to a recent estimate from the WHO, around 25.3 million people in Saudi Arabia (approximately 71.6% of the total population) have received two doses of the COVID-19 vaccine [22]. The vaccination campaign against COVID-19 was initiated in December 2020 in Saudi Arabia, prioritizing frontline workers, the elderly (above 65 years), the obese (BMI >40) and those with co-morbidities. Subsequently, the vaccine was offered to the general population [23,24,25]. Saudi Arabia is one of the nations with strict COVID-19 vaccine mandates, where the primary doses were mandatory while booster doses were recommended for all adults [26,27]. However, the fourth dose of the COVID-19 vaccine was also made available for immunocompromised patients aged less than 50 years [28]. The COVID-19 vaccination coverage was also extended to children aged >12 years in June 2021 as a part of the “safe return to school” campaign [29,30]. The vaccination program among children aged 5 to 11 years was initiated in January 2022. However, vaccination was not mandatory in this age group but was recommended by the health authorities in Saudi Arabia [31,32]. As caregivers of children, parents are responsible for deciding on vaccine uptake among children. In addition to the vaccine’s safety and effectiveness, parental acceptance of vaccinating their children is a key factor for global immunization success. In this context, it is imperative to ascertain parents’ acceptance of the COVID-19 vaccine as well as the factors leading to vaccine hesitancy. This study investigated the intention to vaccinate children against COVID-19 and the barriers linked with childhood COVID-19 vaccination among parents living in the Al-Jouf province of Saudi Arabia.

## 2. Methodology

### 2.1. Ethical Approval

The current study was approved by the Local Committee on Bioethics (LCBE) at Jouf University (Reference No: 08-05-43). The ethical committee permitted an online survey to minimize the exchange of potential fomites, limit the possibility of transmitting COVID-19 and prevent hesitancy among parents about disclosing their health beliefs and decision-making during interviews. Following a brief introduction of the study, online consent was obtained from all participants. The anonymization of data was opted throughout the analysis.

### 2.2. Study Design, Setting and Population

A cross-sectional (February to March 2022) web-based study was conducted among parents from the Al-Jouf region of Saudi Arabia. The parents were included if they (1) had at least one child younger than 12 years, and (2) were residents of the Jouf region and/or nearby suburbs regardless of their COVID-19 vaccination status. If the participants had children who were more than 12 years old as well as children less than 12 years, they answered the questionnaire by keeping their youngest child in mind.

### 2.3. Sampling Technique and Sample Size Calcuation

The sample size was estimated by using OpenEpi 3.0 software. The estimated prevalence of COVID-19 vaccine acceptance was 40%, based on a recently conducted survey [19]. With a confidence interval of 95%, a relative precision of 20% and a design effect of 2, a sample size of 369 was obtained. By adding an estimated dropout rate of 20%, a final sample size of 442 was estimated. An exponential, non-discriminatory snowball sampling technique was used to collect the data from target population.

### 2.4. Development and Validation of the Study Instrument 

A thorough literature review was conducted to identify the key factors of vaccine hesitancy, as well as acceptance, and a web-based study instrument was developed. The initial draft of the questionnaire was developed in English language. The questionnaire was subsequently translated into Arabic language. The translation accuracy was ensured through forward and backward translation method (Arabic to English, English to Arabic). The questionnaire was distributed in both English and Arabic, allowing the respondents to choose their preferred language. The content validity of the study tool was assessed by a panel of experts at Jouf University comprising health professionals from the colleges of pharmacy and medicine. The reliability analysis of the questionnaire was conducted through a pilot analysis on 30 participants, and Cronbach’s alpha was set at 0.78 [7,33]. The study instrument had 3 sections.

Section I consisted of 9 items related to demographics. Apart from basic demographic questions, the participants were asked about their COVID-19 vaccination status and whether their child had received any seasonal influenza vaccine in the past 2 years. The participants were told to answer these questions by keeping the youngest child in mind.Section II consisted of 6 items assessing the participants’ intentions to vaccinate their child with the COVID-19 vaccine. The responses were recorded on a 5-point Likert Scale (very likely, somewhat likely, unsure, somewhat unlikely, very unlikely). For further analysis, these responses were collapsed and recoded with the values for very likely and likely coded as “1”, and the values for very unlikely, unlikely and unsure coded as “0”. However, reverse coding was used for Item 6. The maximum intention score was 6. The mean intention score was estimated. The percentage of agreement was calculated for the participants who agreed to the statements. These statements were ranked accordingly.Section III assessed the barriers to childhood COVID-19 vaccination. This section contained 7 items measuring the barriers on a 5-point Likert (strongly agree to strongly disagree).

### 2.5. Data Collection

A list of parents was initially prepared with the help of students at Jouf University. The initial participants were encouraged to assist in identifying the other participants. These participants were requested to further roll out a survey to eligible candidates. The survey link directed the participants to the informed consent page, followed by the questionnaire. The purpose of the study was also described before the questionnaire. The survey was also distributed to regional groups through social media applications (WhatsApp and Facebook). All the data were transferred to spreadsheets for cleaning and subsequently imported into SPSS for analysis. Data cleaning and import were performed in duplicates (Y.H.K, T.H.M) to ensure the accuracy. A flow diagram of the current study is presented in Figure 1.

### 2.6. Statistical Analysis

The SPSS version 25 was used for data analysis. Descriptive statistics were performed to generate a summary of the study variables as well as the responses of the participants towards the items. Continuous data were expressed as the mean and standard deviation (SD), while categorical data were summarized as the frequency with proportions (%). Chi-square test or Fisher’s exact test for used to compare the categorical data, where appropriate (if the cell count exceeded 5%). The relative importance index (RII) was used to define the relevance of the study tool statements. The RII identified and ranked COVID-19 vaccination intentions as well as the barriers to vaccine uptake. Statements portraying the intentions and barriers to COVID-19 vaccine uptake were ranked to determine the major intention and barrier statements regarding vaccine uptake. The highest value of RII (0 ≥ RII ≤ 1) related to the highest for the major intention and barrier. The RII value was calculated by using the equation
(1)$$\mathrm{RII}=\frac{\sum W}{\left(A*N\right)}$$
where N denotes number of respondents, A represents the highest weight and W is the weight given to each statement by participant (1: strongly disagree; 5: strongly agree). The percentage agreement against each statement was estimated for each participant. To evaluate the percentage of agreement with the statements evaluating the intentions to have the child vaccinated, the 5-point Likert scale was collapsed to 2 points, where “agree and strongly agree became “intending”, while “disagree” and “strongly disagree” became “non-intending”. Comparisons between demographic variables and vaccine uptake intentions were made by using *t*-tests (for continuous variables) or chi-square tests (for categorical variables), where appropriate. Furthermore, factors associated with parental intention to vaccinate their children against COVID-19 were determined by performing binary logistic regression. A series of univariate analyses were performed to assess all the covariates significantly (*p* < 0.05) associated with the dependent variable (parental intention to vaccinate). Subsequently, these statistically significant covariates (*p* < 0.05) were used in the multivariate analysis (method: Enter) to confirm the factors associated with parental intention to vaccinate. The predictive capacity of the regression model was evaluated by the Hosmer–Lemeshow test. A *p*-value of < 0.05 was considered statistically significant throughout the analysis.

## 3. Results

In total, 588 participants responded to our survey and 477 provided consent (participation rate: 477/588 = 81%). Of the 477 received surveys, 33 responses were incomplete and therefore excluded (completion rate: 477/477 = 100%; completeness rate: 444/477 = 93%). This study included 444 parents (mean age: 28.41 ± 7.4 years) with a preponderance of females (65%). Most of the study participants (79.1%) belonged to non-healthcare professions. More than 90% of participants had received the COVID-19 vaccine (two primary doses + one booster dose). A sizeable proportion of parents (41%) reported that their children had received the influenza vaccine in the previous year. Half of the participants (56%) agreed that they made healthcare decisions for their children mutually, while mothers alone (19%) were least responsible for such decisions. The demographic profile of the study participants is summarized in Table 1.

The parents’ intention/willingness towards the childhood COVID-19 vaccine was estimated through six items. The mean intention score (IS) among the study participants was 3.0 ± 1.37. The respondents were stratified as intending (IS > 3) and non-intending (IS ≤ 3) on the basis of their intention score. Only 42.3% (*n* = 188/444) parents intended to vaccinate their children against COVID-19. Table 1 indicates the comparison of parent’s demographics between who intended to vaccinate their children against COVID-19 and those who did not. 

### 3.1. Analysis of Items on Parents’ Intention towards the Childhood COVID-19 Vaccination

The parents’ intention to vaccinate their kids against COVID-19 was determined by six items in the questionnaire. Of these, government mandates for childhood vaccination were ranked top (RII = 0.76, 73%) (Table 2). The statement “I plan to have my child vaccinated but I will wait for some time and see how other children respond to the COVID-19 vaccine” was ranked second highest, with a total of 71% agreement by the respondents. The intention statement “I plan to have my child vaccinated as soon as a vaccine is available for children” was ranked 5th on RII and approximately 21% of participants agreed to it. Unfortunately, only 37% parents were fully committed to vaccinate their kids against COVID-19. Our analysis showed that government mandates, feedback from parents who had vaccinated their children and healthcare professionals’ recommendations were the major drivers of childhood COVID-19 vaccination.

### 3.2. Potential Barriers Associated with the Uptake of the Childhood COVID-19 Vaccination

The current study analyzed seven potential barriers to childhood COVID-19 vaccine uptake (Table 3). Of these, concerns over the vaccine’s safety or side effects were ranked highest (RII: 0.754), reported by 290 (65.3%) participants. The lack of scientific data regarding the efficacy of vaccine in children was ranked second (RII: 0.736), where 62% of participants reported that there is a lack of scientific data regarding the vaccine’s efficacy for children. Approximately half of the participants (48.4%) agreed with the statement that “Availability of too many vaccines makes it difficult to choose the best vaccine for children,” and this barrier ranked third (RII: 0.646). However, the lowest-ranked barrier was religious beliefs about avoiding vaccines, and only 24% of the participants agreed with it. The most frequently reported intention and barrier statements linked with childhood COVID-19 vaccination are presented in Figure 2.

It was interesting to note that 78.2% of parents indicated at least one barrier in this study. The reporting of barriers was more profound among parents who did not intend to vaccinate their children as compared with those who intended to do so (83.6% versus 70.7%, *p* value: 0.001). However, 21.8% of participants did not report any barriers.

### 3.3. Factors Associated with Intention to Childhood COVID-19 Vaccination

As shown in Table 4, fifteen variables were subjected to univariate analysis. Of these nine independent variables were found to be significantly associated with dependent variable in the univariate analysis (parents’ education level, profession, COVID-19 vaccination status, their involvement in children’s healthcare decision, efficacy and safety concerns, family pressure to avoid the COVID-19 vaccine, the belief children were not at risk of the infection (low perceived vulnerability) and the belief that conventional prevention measures were enough to protect children from the infection). Age, gender, current marital status, history of influenza vaccination in children, concerns over the availability of too many vaccines, and religious beliefs against the COVID-19 vaccine were not found to be significantly associated with childhood COVID-19 vaccination intention. Consequently, the significant covariates from univariate analysis were simultaneously entered into the model. The results indicated that our model was statistically significant (χ2 (11) = 74.744, *p* < 0.001) and had good predictive capability (Hosmer–Lameshow test: χ2 (8) = 7.051, *p* = 0.531). Of all the predictors in the model, only four were found to be significant predictors of parental intentions to vaccinate children against COVID-19 (Table 4). Increased parental education level was linked with a positive intention to childhood COVID-19 vaccination, keeping all other covariates constant (secondary education: OR = 3.617, *p* = 0.010; tertiary education: OR = 2.775, *p* = 0.042). Furthermore, parents who had received COVID-19 vaccines were seven times more likely to vaccinate their kids against COVID-19, keeping other independent variables constant. Additionally, the mother’s involvement in decisions about the child’s healthcare (OR 4.353, *p* < 0.001) as well as the father’s and mother mutual decision (OR 3.195, *p* < 0.001) were associated with a higher likelihood of parental COVID-19 vaccine intention. Last but not the least, parents’ trust in the COVID-19 vaccine’s safety was a significant factor for pediatric COVID-19 vaccination (OR 2.483, *p* = 0.022).

## 4. Discussion

This study quantifies parents’ intentions and concerns about the childhood COVID-19 vaccination in Saudi Arabia. It is pertinent to mention that, despite the health authorities’ recommendation for childhood COVID-19 vaccination, only 42.3% of the parents intended to vaccinate their children. The majority of the parents intended to have their children vaccinated if mandated by the Ministry of Health (MOH) in Saudi Arabia. Concerns about the safety and efficacy of COVID-19 vaccines were the most frequently reported barriers to childhood vaccination. According to multivariate logistic regression analysis, education level, the parents’ vaccination status, the mother’s sole or mutual decision-making regarding the child’s health, and the parents’ trust in the safety of COVID-19 vaccines were found to be significantly associated with pediatric COVID-19 vaccination intention. These findings carry pivotal implications to improve the vaccination coverage.

The achievement of herd immunity through massive immunization has been a global goal. The Kingdom of Saudi Arabia has, so far, achieved a 70% adult vaccination rate, with approximately 69 million doses administered by November 2022 [21,34]. Approximately one-third of the Saudi population is less than 18 years old [35]. The inclusion of this population into the vaccination program is of utmost importance. To achieve herd immunity and the normalization of daily activities, the Saudi government initiated compulsory vaccination of those aged between 12 and 18 years in 2021 [29], followed by vaccination of children aged 5 to 12 years at the start of the year 2022 [32].

The intention rate among parents reported in our study is comparatively lower than the overall proportion estimated by a systematic review that synthesized the data of 43 studies from 18 countries. This data synthesis indicated that parents’ intentions towards childhood COVID-19 vaccination ranged from 25.6% to 92.2%, where the pooled incidence rate was 60.1% with a heterogeneity of 99.9% [36]. The high value of heterogeneity in this systematic review indicates disparities in outcomes assessment. This wide variation in the intention rate can be explained by the disparity in the quality of the studies, the method of recruiting parents, the timeframe of data collection and the study sites. However, it is important to note that this review included four studies on the Saudi population, where the intention rates ranged from 25.6% to 54.1% [17,37,38,39]. It is worth mentioning that parental vaccine hesitancy was prevalent in Saudi Arabia even before the COVID-19 pandemic [18]. Recent investigations in Saudi Arabia have also reported considerable hesitancy against COVID-19 vaccines among the general population, regardless of marital status [40,41]. Al-Mohaithef et al. surveyed 658 participants from the general public and found that only 53% of the study population intended to receive the COVID-19 vaccine. The results of multivariate analysis showed that trust in the healthcare system was a major factor that affected the participants’ acceptance of childhood vaccination (OR: 3.4, 95% CI) [41]. Since the hesitant behavior of the adult population toward vaccination can impact childhood vaccination coverage, addressing the parents’ hesitancy is a key to success in overall vaccination campaigns.

The childhood COVID-19 vaccination intention rate reported in our study (42.3%) is in concordance with various national and international surveys [39,42,43,44]. Al-Musbah et al. conducted a study among Saudi parents and indicated that only 28.1% of parents agreed to vaccinate their children against COVID-19 [38]. The low acceptance in their study might be associated with data collection time and sample size. The authors collected the data during May–June 2021, when the vaccination campaign for children had not been initiated, and most of the adult population had only received one dose. Ennaceur et al. reported intention rate of 44% among Saudi parents for COVID-19 vaccination of their children [42]. On the other hand, Al-Mansour et al. reported a comparatively higher proportion (57.8%) of COVID-19 vaccine acceptance among Saudi parents [45]. This disparity in the intention rate might be explained by the variations in the study populations and sites, as Al-Mansour and colleagues collected their data primarily from metropolitan regions where the number of COVID-19 cases was higher compared with our study site. It is pertinent to mention that parents’ intentions to vaccinate their children are subject to change due to the evolving data on the vaccine’s efficacy and safety. This phenomenon explains the variations in the reported intention rates across the studies. It is possible to have variations in the intention rate if the same population is surveyed at two different time points. Previous investigations have indicated that vaccine hesitancy is not a stable but a labile trait, and can transition to acceptance or rejection of the vaccine [46,47]. In this context, it is important to evaluate the behavior of the general population toward vaccination at various time points.

This study reported various factors associated with parents’ intention to vaccinate their children. These factors are aligned with most of the previous studies [42,43,44,48]. Ennaceur et al. reported that parents with higher likelihood to vaccinate themselves (OR: 0.599, 95% CI: 0.367–0.980) and who showed trust in the healthcare system (OR: 0.527, 95% CI: 0.327–0.848) demonstrated a higher acceptance of children’s vaccination [42]. Similar findings have been observed in our study, where parents vaccinated against COVID-19 were 7.1 times more likely to vaccinate their children. For any health-related problem, the requirement for trustworthy information and reliance on the sources that offer it are of utmost importance. In the instance of COVID-19, where information and counter-information were constantly changing, it is very crucial to have a reliable information source. This result is in line with earlier studies, where the majority (86%) of the parents turned to the Saudi MOH for information about COVID-19 [39]. These figures emphasize the effectiveness of the local government in encouraging immunization among the local population. Tailored vaccination awareness programs would help to increase parents’ compliance with vaccination. Furthermore, addressing parents’ concerns about childhood COVID-19 vaccination will aid in maintaining the vaccination coverage graph.

Another factor that increases parents’ intentions toward COVID-19 vaccination is the response or experience of other children with the vaccine. A similar trend has been observed with influenza vaccinations. Existing evidence indicates that the parents who received an influenza vaccine were more likely to accept COVID-19 vaccines [49]. However, despite 41% (182/444) of children already having been vaccinated against influenza in our study, a total of 52% (96/182) refused to vaccinate their children against COVID-19 (Table 1). This might be due to the low vaccination rate of children in the northern region as compared with the western region of KSA. A study by Ghada et al. showed that 23% had delayed pediatric vaccination in Sakaka, KSA, compared with only 9% delaying vaccination in Jeddah, a metropolitan city of KSA [20,50]. Another positive factor that increases parents’ intentions is advice from healthcare professionals. Healthcare professionals constantly act as ambassadors to encourage vaccination acceptance. During the COVID-19 pandemic, the importance of their responsibilities increased. However, the rate of parental acceptance of the COVID-19 vaccination was substantially lower (25%) among parents belonging to the healthcare professions in our study. It might be because healthcare professionals are better informed than the general public regarding the low severity of COVID-19 among children, along with the fact that children are at minimal risk of dying from COVID-19 and that the majority of infected children would not exhibit severe symptoms. However, the low perceived severity of and vulnerability to COVID-19 among children were also prevalent among non-health professionals [43]. This was also evident in our study, where around 40% of parents demonstrated a low perceived vulnerability to COVID-19 in children. Since the low perceived susceptibility of contracting COVID-19 by children can undermine the ongoing maneuvers of the vaccination programs, the health authorities should initiate educational campaigns for parents.

Our study contributes to the body of existing literature by underscoring the characteristics of parents who might be reluctant to pursue childhood COVID-19 vaccination. Overall, the results demonstrated that parents’ intentions to have their children vaccinated against COVID-19, in the end, depend mainly on their educational level and vaccination status against COVID-19, along with which parent had responsibility for making health-related decisions regarding the child. Interestingly, the family pattern of decision-making for a child’s health was found to be associated with vaccination intentions in our study. This study showed that the involvement of mothers and mutual involvement (father and mother) in making decisions for the child’s health were linked with a higher likelihood of vaccinating children against COVID-19. A recent investigation has indicated that females perceive themselves to be at a greater risk of contracting COVID-19 and thus have a higher likelihood of receiving vaccines [51]. However, the role of gender in vaccine acceptance for children seems inconclusive due to variations in social and cultural norms. The existing literature does not provide a firm conclusion, as one meta-analysis endorses a higher vaccination rate among females [52], while another meta-analysis indicates more vaccination acceptance among males than in females [53]. These results necessitate the need for further investigations.

It is critical to comprehend both the elements that count and those that do not when parents are making judgments about whether to vaccinate their children against COVID-19. The barriers were assessed via seven statements covering the following themes: safety/efficacy of the vaccines, availability of COVID-19 vaccines, perceived seriousness of the disease, perceived risk of the disease, and social and religious pressure to avoid COVID-19 vaccines. The current study proposed vaccine safety (side effects) and efficacy (lack of scientific data) as major barriers to COVID-19 vaccines. These findings are consistent with recently published studies where parents expressed concerns about the side effects of adult immunizations and expressed skepticism about administering the same vaccines to their children [42,54]. Additionally, many parents felt that there have been too many vaccines introduced in a short period and there is a lack of scientific evidence addressing the long-term effects of immunization among children [54]. The foundation of many health behavior theories lies in the sense of threat of the disease. According to a meta-analysis, vaccine reluctance is highly correlated with the level of risk, susceptibility and severity of the disease [49]. Although disease susceptibility and severity were not highly rated barriers in our study, many parents agreed that wearing masks and practicing social distancing were enough to protect children without the need for vaccination. Contrary to the belief that many parents avoided vaccination in children for religious reasons, the parents in our study reported religious causes as the lowest ranked barrier to childhood COVID-19 vaccination. Finally, the logistic regression analysis indicated that the parents who believed in the safety profile of vaccines had higher odds of vaccinating their children against COVID-19. These findings corroborate the results of other studies [42,43,44,45,48].

This study is accompanied by few shortcomings including small sample size, the potential generalizability and timeframe, which should be taken into consideration when analyzing the results. Furthermore, as we did not use cookies or collect IP addresses to identify unique visitors to the first page of our survey, we could not estimate the true response rates (view rates and participation rates). Future studies should consider the E-survey checklist to ensure the quality of web surveys [55]. It is pertinent to mention that our study looked at intentions and barriers to the COVID-19 vaccine in February 2022, when the government of KSA had just offered vaccination for children aged 5–12 years. The data we have gathered may not accurately reflect current parental vaccination decisions (November 2022); instead, they show parental plans to vaccinate their child as of February 2022. Moreover, while we acknowledge the value of national samples, there are also particular advantages to representing household samples in densely populated remote areas, which allow for a more in-depth examination of research questions about the respondents’ individual local environmental and cultural contexts. Although the burden of COVID-19 sickness and mortality has been concentrated in several metropolitan areas across the country during the pandemic, we expect that the findings from our sample of households is likely to be highly relevant to other less developed areas around the country with high disease transmission and low vaccination rates. Moreover, this study is strengthened by a detailed elaboration of factors related to intention and barriers to childhood COVID-19 vaccination. The findings originated from this study may assist in devising policies and vaccination campaigns for childhood vaccination against COVID-19.

## 5. Conclusions

This study indicated that even in a country with high parental vaccine uptake, there might be substantial hesitancy towards the vaccination of young children against COVID-19. The reasons for hesitancy about the COVID-19 vaccine in children are complex, with vaccine side effects as the major barrier. The health authorities should focus on addressing parental concerns about vaccines and make sure that parents have an adequate understanding of the importance of vaccination. These maneuvers would aid in improving childhood vaccination against COVID-19. Furthermore, awareness about COVID-19 vaccines and confidence in the health authorities constitute background factors that predispose parents towards vaccinations.

## Figures and Tables

**Figure 1 vaccines-10-02093-f001:**
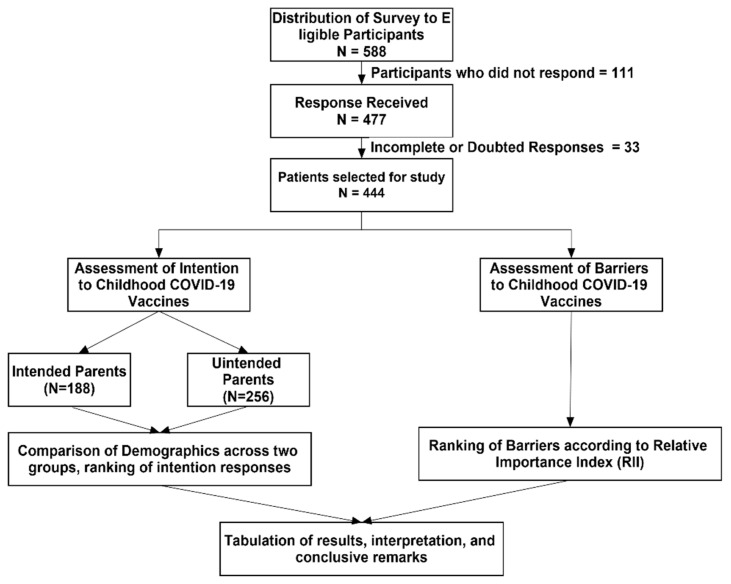
Methodological flowchart of the study.

**Figure 2 vaccines-10-02093-f002:**
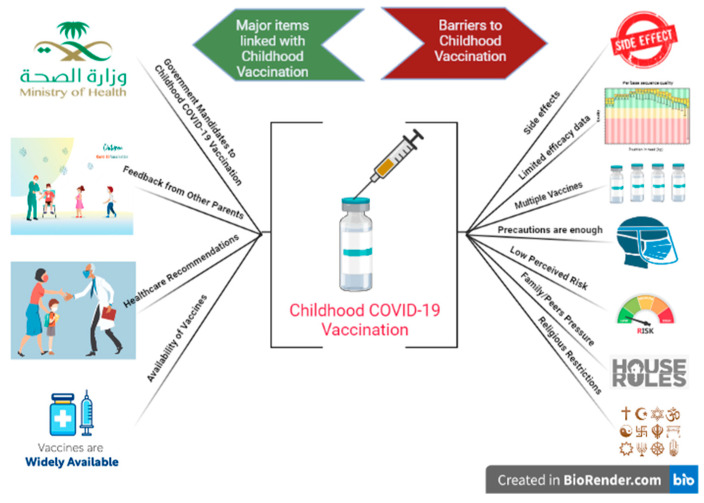
Themes related to the intentions and barriers of childhood COVID-19 vaccination generated by this study.

**Table 1 vaccines-10-02093-t001:** Demographic features of parents included in this study.

Demographic Variable	Total Participants (*n* = 444)	Intending * (*n* = 188)	Non-Intending ** (*n* = 256)	*p*-Value
Age (Mean ± SD); Years	28.41 ± 7.4	28.55 ± 7.6	28.30 ± 7.2	0.734
Gender				
Male	155 (34.9%)	56 (29.8%)	99 (38.7%)	0.052
Female	289 (65.1%)	132 (70.2%)	157 (61.3%)	
Marital status				
Married	292 (65.8%)	128 (68.1%)	164 (64.1%)	0.524
Divorced	79 (17.8%)	29 (15.4%)	50 (19.5%)	
Separated	73 (16.4%)	31 (16.5%)	42 (16.4%)	
Educational level				
Preparatory	38 (8.6%)	6 (3.2%)	32 (12.5%)	0.001
Secondary	197 (44.3%)	80 (42.6%)	117 (45.7%)	
University	209 (47.1%)	102 (54.2%)	107 (41.8%)	
Profession				
Healthcare	93 (20.9%)	27 (14.4%)	66 (25.8)	0.003
Non-healthcare	351 (79.1%)	161 (85.6%)	190 (74.2%)	
Received the COVID-19 vaccination				
Yes	402 (90.5%)	185 (98.4%)	217 (84.8)	<0.001
No	42 (9.5%)	3 (1.6%)	39 (15.2%)	
Child received an influenza vaccination in the past year				
Yes	182 (41.0%)	86 (45.7%)	96 (37.5%)	0.081
No	262 (59.0%)	102 (54.3%)	160 (62.5%)	
Who makes the healthcare decisions for the children				
Father	84 (18.9%)	20 (10.6%)	64 (25.0%)	<0.001
Mother	110 (24.8%)	55 (29.3%)	55 (21.5%)	
Both	250 (56.3%)	113 (60.1%)	137 (53.5%)	

SD: standard deviation. Two groups were compared by using student’s *t*-tests and χ^2^ tests. ***** Participants who intend to have their children vaccinated. ** Participants who do not intend to vaccinate their children.

**Table 2 vaccines-10-02093-t002:** Intention to have children vaccinated against COVID-19.

Statement	Very Likely *n* (%)	Somewhat Likely *n* (%)	Unsure *n* (%)	Somewhat Unlikely *n* (%)	Very Unlikely *n* (%)	RII *	Rank ^£^	% Agreement ^€^
I plan to have my child vaccinated against COVID-19.	68 (15.3)	95 (21.4)	42 (9.5)	108 (24.3)	131 (29.5)	0.485	6	163 (36.7%)
I plan to have my child vaccinated as soon as a vaccine is available for children.	7 (1.6)	87 (19.6)	130 (29.3)	189 (42.6)	31 (7)	0.532	5	94 (21.2%)
I plan to have my child vaccinated but I will wait for some time and see how other children respond to the COVID-19 vaccine.	147 (33.1)	169 (38.1)	45 (10.1)	24 (5.4)	59 (13.3)	0.745	2	316 (71.2%)
I will vaccinate my children if it is made compulsory for schools, daycare, restaurants and malls by the government.	183 (41.2)	144 (32.4)	22 (5)	36 (8.1)	59 (13.3)	0.760	1	327 (73.6%)
I will vaccinate my children if their doctor/healthcare provider recommends it.	173 (38.9)	120 (27)	39 (8.8)	46 (10.4)	66 (14.9)	0.730	3	293 (66%)
I do not plan to vaccinate my children.	95 (21.4)	97 (21.8)	114 (25.7)	56 (12.6)	82 (18.5)	0.630	4	192 (432%)

* Relative importance index. ^£^ The ranking of items according to the RII score. ^€^ Percentage of respondents who agreed to the statement (combined response: very likely + somewhat likely).

**Table 3 vaccines-10-02093-t003:** Potential barriers to uptake of the childhood COVID-19 vaccination.

Statement	Strongly Agree	Agree	Neutral	Strongly Disagree	Disagree	RII *	Rank ^£^	% Agreement ^€^
Lack of scientific data regarding the vaccine’s efficacy for children.	132 (29.7)	143 (32.2)	77 (17.3)	78 (17.6)	14 (3.2)	0.736	2	275 (61.9%)
Concerns about the vaccine’s safety/side effects.	154 (34.7)	136 (30.6)	70 (15.8)	67 (15.1)	17 (3.8)	0.754	1	290 (65.3%)
The availability of too many vaccines makes it difficult to choose the best vaccine for children.	82 (18.5)	133 (30)	82 (18.5)	99 (22.3)	48 (10.8)	0.646	3	215 (48.4%)
Religious beliefs about avoiding the vaccine.	38 (8.6)	68 (15.3)	135 (30.4)	134 (30.2)	69 (15.5)	0.542	7	106 (23.9%)
Family/peer pressure to avoid the vaccine.	60 (13.5)	100 (22.5)	89 (20.1)	134 (30.2)	61 (13.7)	0.584	6	160 (36%)
Children are not at risk of COVID-19 complications.	99 (22.3)	84 (18.9)	81 (18.2)	127 (28.7)	53 (11.9)	0.622	5	183 (41.2%)
Wearing masks, using sanitizers and practicing social distancing are enough to protect children.	60 (13.5)	147 (33.1)	78 (17.6)	102 (23)	57 (12.8)	0.623	4	207 (46.6%)

* Relative importance index. ^£^ The ranking of items according to the RII score. ^€^ Percentage of respondents who agreed to the statement (combined response: strongly agree + agree).

**Table 4 vaccines-10-02093-t004:** Factors associated with Intention to Childhood COVID-19 Vaccination.

Covariates	Univariate Analysis	*p*-Value	Multivariate Analysis	*p*-Value
	COR (95% CI)		AOR (95% CI)	
Age	1.004 (0.979–1.030)	0.733		
Gender				
Male	1.00 (Ref.)			
Female	1.486 (0.995–2.220)	0.053		
Marital status				
Married	1.00 (Ref.)			
Divorced/separated	0.836 (0.561–1.245)	0.378		
Educational level				
Preparatory	1.00 (Ref.)		1.00 (Ref.)	--
Secondary	5.084 (2.040–12.670)	<0.001	3.617 (1.353–9.672)	0.010
University	3.647 (1.457–9.125)	0.006	2.775 (1.037–7.429)	0.042
Profession				
Healthcare	1.00 (Ref.)		1.00 (Ref.)	
Non-healthcare	2.071 (1.263–3.396)	0.004	1.390 (0.783–2.468)	0.261
Received the COVID-19 vaccination				
No	1.00 (Ref.)		1.00 (Ref.)	--
Yes	11.083 (3.370–36.451)	<0.001	7.062 (1.919–25.991)	0.003
Child received the influenza vaccination in the past year				
No	1.00 (Ref.)			
Yes	1.405 (0.959–2.060)	0.081		
Who makes the healthcare decision for the children?				
Father	1.00 (Ref.)		1.00 (Ref.)	--
Mother	3.200 (1.711–5.984)	<0.001	4.353 (2.164–8.756)	<0.001
Both mother and father (mutual)	2.639 (1.507–4.623)	0.001	3.195 (1.696–6.028)	<0.001
Lack of scientific data regarding the vaccine’s efficacy for children				
Yes	1.00 (Ref.)		1.00 (Ref.)	
No	2.593 (1.619–4.152)	<0.001	1.608 (0.793–3.260)	0.188
Concerns about the vaccine’s safety in children				
Yes	1.00 (Ref.)		1.00 (Ref.)	
No	3.036 (1.851–4.979)	<0.001	2.483 (1.143–5.392)	0.022
The availability of too many vaccines makes it difficult to decide which one is better for children				
Yes	1.00 (Ref.)			
No	0.837 (0.560–1.252)	0.387		
Religious beliefs against vaccination				
Yes	1.00 (Ref.)			
No	1.453 (0.995–2.122)	0.053		
Family/peer pressure to avoid childhood COVID-19 vaccination				
Yes	1.00 (Ref.)		1.00 (Ref.)	
No	1.719 (1.174–2.516)	0.005	0.983 (0.559–1.726)	0.952
Belief that children are not at risk of COVID-19				
Yes	1.00 (Ref.)		1.00 (Ref.)	
No	1.510 (1.029–2.215)	0.035	0.690 (0.375–1.271)	0.234
Wearing masks, using sanitizers and practicing social distancing are enough to protect children				
Yes	1.00 (Ref.)		1.00 (Ref.)	
No	1.660 (1.121–2.457)	0.011	1.012 (0.551–1.860)	0.969

AOR, adjusted odds ratio; CI, confidence interval; COR, crude odds ratio.

## Data Availability

All the data underlying the findings of this study are present in this manuscript.

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
