# Peer review of "Parental Perceptions and Barriers towards Childhood COVID-19 Vaccination in Saudi Arabia: A Cross-Sectional Analysis"

_vaccines, 2022, doi:10.3390/vaccines10122093_

Round 1

Reviewer 1 Report

In my opinion the paper was not clear in a satisfactory manner, readable and informative and will not provide a valuable source document for anyone requiring a primer to know and understand this issue. Numerous shortcomings in the sections Introduction, Materials/Methods, Results and Discussion make this paper inadequate for publishing in its current form. If the authors are interested, some comments below could be of use to them in the future. Some comments:    
  • Lines 2-5: Shorten the title of the paper, it is unnecessary to mention twice `COVID-19`. 
  • Line 17: When first mentioning the abbreviation (in Abstract and entire manuscript), instead of `COVID-19` (and any other abbreviation) inscribe the full name. 
  • Lines 31-32: In this manuscript the results of analysis which would allow the conclusion `The vaccination intention was found to be associated with education, profession and vaccination status of the parents.` are not stated at all. This manuscript gives a description of participants  of the study and assessment of the significance of differences between 2 groups of participants. But, an association between `Parent`s Intention to Vaccinate their Children against COVID-19 and Barriers Associated with Childhood COVID-19 Vaccination` and demographic variables is not presented in this manuscript.    
  • Lines 41-43: Check and correct the numbers stated in this sentence listed for persons who are `fully vaccinated` and persons who are `partially vaccinated`. It would be useful for readers to state in this sentence that these are estimates by the World Health Organization, when referring to pandemic. Cite correctly in the text of the paper and in the list of References the stated reference No 1. 
  • Lines 44-46: The data stated in this sentence are not in the cited reference No 2. The cited reference does not at all mention either adolescents or children, but on the contrary refers to older adults (70+ years). Also, in this reference neither Canada or the USA are mentioned. Cite the appropriate reference which will confirm the information stated in this sentence. 
  • Lines 48-50: Cite the appropriate reference in this sentence. 
  • Lines 50-52: Cite the appropriate reference in this sentence. 
  • Lines 52-56: Cite more precisely the results presented in all three cited references (references 4-6): instead of `adolescents` state the exact age which is described in cited references. Also, avoid the word `relatively` by stating the exact frequency of side effects.  
  • lines 63-64: The reference No 12 is inappropriately cited. Since you are stating that `the World Health Organization (WHO) has listed it as one of the major global health threats`, you should therefore cite an adequate reference which refers to WHO.  
  • Lines 66-67: Define the abbreviation `KSA` in this sentence. 
  • Lines 71-73: Add data regarding the age for which vaccination against COVID-19 is mandatory, and the age for which it is recommended, in Saudi Arabia. Additionally, state the coverage of vaccination against COVID-19 in Saudi Arabia, with citation of an appropriate reference. State what is the status of vaccination against COVID-19 among children in Saudi Arabia: for which age and whether it is mandatory or recommended vaccination, and since when. Cite an appropriate reference.  
  • Line 104. Add `Participation Rate` and `Response Rate` in this manuscript. 
  • lines 130-131: To which section does the information for Cronbach`s alpha refer to? Cite the reference which presents the results of the assessment of psychometric characteristics of the used questionnaire. 
  • Lines 165-170: In this manuscript the results of statistical analyses listed in these sentences are not presented. Explain.
  • Lines 170-171: Is it correctly stated in this sentence that `P-value of < 0.005 was considered statistically significant.`?  
  • Lines 187-188: On Table 1 pay attention to the distribution of participants which is expressed in percentages (%), the sum of participants should probably be 100.0% for all presented variables, and align the results. Also, align the number of decimals for % (see: Father 25%). The comment goes for all Tables in this manuscript.  
  • Line 215: The description of Figure 2 is missing.  
  • Lines 223-279: Reconstruct the entire section Discussion (references are not listed in an order, e.g. reference No 28 is cited and then reference No 18, etc.). In the entire text of the paper references 19-26 are not cited at all, while they are listed in the list of References.  
  • Line 237: The abbreviation MOH is mentioned only once in this paper, inscribe its meaning.
  • Lines 318-382: List of References should be in line with Instructions to authors.  

Author Response

Point-by-Point response to reviewers

Respected Editor,

We have received revisions/suggestions for our submitted manuscript. All the concerns and suggestions of the reviewers have been addressed and we hope that the revised version of the manuscript will satisfy the concerns of all reviewers. We have attached a point-by-point response to each reviewer and a revised version of the manuscript for your consideration. Please let us know if any other changes are required in this regard. Last but not least, we are very thankful to the editor and all reviewers for their time and efforts in put their valuable suggestions. Indeed, their recommendations made this manuscript more scientifically elegant and sound.

Reviewer 1

General Query: In my opinion the paper was not clear in a satisfactory manner, readable and informative and will not provide a valuable source document for anyone requiring a primer to know and understand this issue. Numerous shortcomings in the sections Introduction, Materials/Methods, Results and Discussion make this paper inadequate for publishing in its current form. If the authors are interested, some comments below could be of use to them in the future.

Response: Respected reviewer, we are thankful to you for your time and comments. We always welcome critics of our work as they open several opportunities to learn and improve. In this context, we welcome all your suggestions and incorporated them in the revised version of the manuscript. We will also welcome further suggestions in this regard.

We have tried our level best to address your comments and incorporate changes in our manuscript accordingly. We confirm that these suggestions have improved our manuscript in terms of its readability and scientific soundness. Again we would like to thank you for these comments and request you to let us know if you want further modifications.

Query 1: Lines 2-5: Shorten the title of the paper, it is unnecessary to mention twice `COVID-19`.

Response 1: Respected reviewer, the title of paper has been changed to following

“Parental Perceptions and Barriers towards Childhood COVID-19 vaccine in Saudi Arabia: A Cross-sectional analysis”

Query 2: Line 17: When first mentioning the abbreviation (in Abstract and entire manuscript), instead of `COVID-19` (and any other abbreviation) inscribe the full name.

Response 2: Respected reviewer, as per your recommendation, full inscriptions of abbreviations are added throughout the manuscript and highlighted.

Query 3: Lines 31-32: In this manuscript the results of analysis which would allow the conclusion `The vaccination intention was found to be associated with education, profession and vaccination status of the parents. ` are not stated at all. This manuscript gives a description of participants in the study and assessment of the significance of differences between 2 groups of participants. But, an association between `Parent`s Intention to Vaccinate their Children against COVID-19 and Barriers Associated with Childhood COVID-19 Vaccination` and demographic variables is not presented in this manuscript.

Response 3: Respected reviewer, Table 1 indicates the association of demographics with intention to vaccinate kids. The association or relationship of demographics with vaccine intention was determined by Chi-square test.

Query 4 Lines 41-43: Check and correct the numbers stated in this sentence listed for persons who are `fully vaccinated` and persons who are `partially vaccinated`. It would be useful for readers to state in this sentence that these are estimates by the World Health Organization, when referring to pandemic. Cite correctly in the text of the paper and in the list of References the stated reference No 1.

Response 4: Respected reviewer, we have changed “According to World Health Organization” to give readers a clear reference.  Moreover, we have corrected the reference.

Query 5: Lines 44-46: The data stated in this sentence are not in the cited reference No 2. The cited reference does not at all mention either adolescents or children, but on the contrary refers to older adults (70+ years). Also, in this reference neither Canada or the USA are mentioned. Cite the appropriate reference which will confirm the information stated in this sentence.

Response 5: Respected reviewer, we apologize for this error which probably linked with a malfunction of our endnote software. We have rechecked the references and corrected all, wherever needed. We are really very sorry for the inconvenience. The citation errors occur due to linking the wrong library with our software as some libraries share closely related names. We believe that the revised version will be free of citation errors.

Query 6: Lines 48-50: Cite the appropriate reference in this sentence.

Response 6: Respected reviewer, references have been corrected in the text as well as bibliography at the end with reference number 4,5.

Query 7: Lines 50-52: Cite the appropriate reference in this sentence.

Response 7: Respected reviewer, reference has been corrected in the text as well as bibliography with reference number 6.

Query 8: Lines 52-56: Cite more precisely the results presented in all three cited references (references 4-6): instead of `adolescents` state the exact age which is described in cited references. Also, avoid the word `relatively` by stating the exact frequency of side effects. 

Response 8: Respected reviewer, we have mentioned the age, as well as results of these 3 references. We hope that this precision will satisfy the concern.

Query 9: lines 63-64: The reference No 12 is inappropriately cited. Since you are stating that `the World Health Organization (WHO) has listed it as one of the major global health threats`, you should therefore cite an adequate reference which refers to WHO. 

Response 9: Respected reviewer, the reference has been replaced that now the new reference number for this sentence is 15.

Query 10: Lines 66-67: Define the abbreviation `KSA` in this sentence.

Response 10: Respected reviewer, the Kingdom of Saudi Arabia has been added. Any abbreviation that has been used for the first time has been checked throughout the manuscript and in-scripted accordingly. Thank you for pointing out this issue in the manuscript.

Query 11: Lines 71-73: Add data regarding the age for which vaccination against COVID-19 is mandatory, and the age for which it is recommended, in Saudi Arabia. Additionally, state the coverage of vaccination against COVID-19 in Saudi Arabia, with citation of an appropriate reference. State what is the status of vaccination against COVID-19 among children in Saudi Arabia: for which age and whether it is mandatory or recommended vaccination, and since when. Cite an appropriate reference.

Response 11: Respected reviewer, the timeline and the status of COVID-19 vaccination has been provided in the last paragraph of introduction section. Moreover, appropriate citations were made from the Ministry of Health and official news agency.  

Query 12: Line 104. Add `Participation Rate` and `Response Rate` in this manuscript.

Response 12: Respected reviewer, the response rate has been mentioned in figure 1, however, based on your recommendation, we have explained it descriptively in manuscript under start of “Result” heading.

Query 13: lines 130-131: To which section does the information for Cronbach`s alpha refer to? Cite the reference which presents the results of the assessment of psychometric characteristics of the questionnaire used.

Response 13: Respected reviewer, Cronbach’s alpha value refers to the intention and barriers part of questionnaire. To test the clarity and comprehensibility of survey content, a pilot study was conducted by including 30 participants who were excluded from the formal evaluation. The Cronbach’s alpha test of internal consistency was used to assess the reliability of the survey tool. The test suggested that the survey tool was reliable overall and Cronbach’s alpha was equal to 0.78, which is above 0.70 as a general cut-off limit. This has been explained in the methodology section under sub-heading 2.4 (Study instrument; Development and Validation). Moreover, references have been added to support psychometric assessment of study tools.

Query 14: Lines 165-170: In this manuscript the results of statistical analyses listed in these sentences are not presented. Explain.

Response 14: Respected reviewer, we have used Kolmogorov–Smirnoff and Shapiro–Wilk tests to check data distribution, mean (continuous variables) and percentages (Categorical variables) for descriptive analysis of data, Chi-square test or the Fisher exact test (for comparison of data), %). The relative importance index (RII) was used to define the relevance of the study tool statements and percentage agreement. Apart from the above-mentioned tests, all the irrelevant information has been removed.

Query 15: Lines 170-171: Is it correctly stated in this sentence that `P-value of < 0.005 was considered statistically significant. `? 

Response 15: Respected reviewer, this is a typo error. A p-value ≤ 0.05 was considered statistically significant. It has been corrected in manuscript.

Query 16: Lines 187-188: On Table 1 pay attention to the distribution of participants which is expressed in percentages (%), the sum of participants should probably be 100.0% for all presented variables, and align the results. Also, align the number of decimals for % (see: Father 25%). The comment goes for all Tables in this manuscript. 

Response 16: Respected reviewer, decimal places has been aligned in all tables. We have cross checked the sum of total participants is 444 in all tables.

Query 17: Line 215: The description of Figure 2 is missing.

Response 17: Respected reviewer, Figure 2 is a pictorial description of the themes that we have used to evaluate intentions and barriers towards childhood COVID-19 vaccination. Since the themes of intentions and barriers are mentioned under result section (table 2 and 3 respectively) we have added pictorial representation in Figure 2 under heading 3.1 and 3.2 respectively in the text of manuscript.

Query 18: Lines 223-279: Reconstruct the entire section Discussion (references are not listed in an order, e.g., reference No 28 is cited and then reference No 18, etc.). In the entire text of the paper references 19-26 are not cited at all, while they are listed in the list of References. 

Response 18: Respected reviewer based on your recommendation and comments received from reviewer 2, many points have been added in the discussion section and therefore new references have been added too. The discussion section is reconstructed. Missed references belonged to the first paragraph of discussion section and have also been added. I hope that the revised version of the manuscript will satisfy the concern

Query 19: Line 237: The abbreviation MOH is mentioned only once in this paper, inscribe its meaning.

Response 19: Respected reviewer, the MOH has been inscribed as Ministry of Health in the manuscript.

Query 20: Lines 318-382: List of References should be in line with Instructions to authors.

Response 20: Respected reviewer, all the references have been made in line with instructions to authors. Thank you for highlighting this issue.

At the end, we all authors are very thankful for your time and efforts paid to review the current manuscript. In addition, we are also impressed with your skills, which provided us with opportunities to learn more. Indeed, your input contributed significant improvements in our draft. We welcome any other suggestions. Thank you.

Dr. Tauqeer Hussain Mallhi, PhD

Reviewer 2 Report

Kindly see the attached file as I can not paste my figures here.

Review of paper ‘Determination of Parent`s Intention to Vaccinate their Children 2 against COVID-19 and Barriers Associated with Childhood 3 COVID-19 Vaccination in Saudi Arabia – A Cross Sectional 4 Analysis’ by Y. H. Khan et al.

This study discussed child vaccination intake against COVID and the parental role in this regard; area of focus is Al-Jouf in Saudi Arabia. Among the sample size, 90% of parents were vaccinated, though only 42% of parents vaccinated their kids. Parental intention and potential barriers were mainly focused. It found a low intention among parents to vaccinate their kids against COVID-19 which depends on the education, profession and vaccination status of the parents. It is useful work and I recommend it after some revision.

Main point:

1.       One issue here could be the sample size as only 444 parents took part in the survey.  Hence it is not a very robust result in terms of sample size. However, I believe results could be very similar if more samples are considered and the survey period is kept the same. Mention in the end that it is a limitation of this study.

2.       If there are comparison with one or two more locations of Saudi Arabia it could strengthen the findings. You can not include that now but at least mention that part at the end also in the discussion section.

3.       Mention the period of vaccination and the survey date of conducting data. This is because Saudi Arabia practically kept all vaccine doses very low since March 2022, compared to the previous period. (https://ourworldindata.org/explorers/coronavirus-data-explorer)

If you do similar analyses now and take the survey, most results will be much different.

Most of the countries worldwide are carefully monitoring vaccine success stories as claimed by the vaccine manufacturer. It is expected that all the observed data are thoroughly investigated and analysed from time to time. As the data are available in the public domain it is easier for the authority to monitor the results easily. Many countries took decisions regarding vaccine doses and  stopped vaccination.  

4.       You may like to mention a few lines about why Saudi Arabia stopped the vaccine booster doses and see the figure below for cumulative doses. (https://ourworldindata.org/explorers/coronavirus-data-explorer).  Add few lines stating why many countries already stopped all vaccine doses including booster (Gibraltar, Iceland among others) or kept the dose nominal (Denmark, Israel etc.) since March 2022. See the trend for Israel and Denmark, which are practically nil since last March. This figure shows like many other countries, Saudi Arabia also stopped booster doses since March 2022. Discuss possible reasons why those countries took such stunts. Many of those countries were very proactive at the initial stages of vaccination.   

5.       You discussed many possible causes of COVID vaccine hesitancy. Here also discuss possible causes of stopping all vaccine doses in many countries.

6.       You mentioned about Herd Immunity (line 225): “To achieve herd immunity and normalization of daily activities, at the end of June 2021, the Saudi government initiated compulsory vaccination of those aged between 12-18 years followed by massive vaccination of aged 5-12 years in February 2022.”

In this regard Africa only had 47 % vaccination, much lower than herd immunity. Cases and deaths are very low. Moreover, Africa reduced vaccine doses drastically in the recent period and cases are showing a rapid decrease. (https://ourworldindata.org/explorers/coronavirus-data-explorer). On the other hand, many highly vaccinated countries those had a much high vaccination rates than the herd immunity threshold, performed much worse in the later period. Include some discussion about those important issues in the discussion.

7.       Flu vaccine comparison in Table 1. The Influenza (Flu) vaccine was never compulsory for children in countries like UK and US. However, Europe and US are worst affected due to seasonal Flu in every winter. I wonder why so many children in Saudi Arabia are vaccinated every year with the Flu vaccine. Kindly give brief statistics of Flu deaths in Saudi Arabia. Elaborate a few lines about Flu season, adult Flu vaccination rate, etc.  

Minor points:

1.       In line 245, Table 4: there is no Table 4 in the text.

2.       Figure 1 legend is not right. Rewrite it.

3.       Line 268: rewrite this line “These results are in concordance with recently published literature where parents are concerned about adult vaccine side effects and are skeptical to use same the vaccines in their children”

4.       I like the point you mentioned about child vaccination in line 249: “However, the rate of parental acceptance of the COVID-19 vaccination was substantially lower (25%) among parents belonging to healthcare professions in our study. It might be because healthcare professionals are more informed than the general public regarding COVID-19 related low severity among children along with the fact that children are at minimal risk of mortality from COVID-19, and that the majority of infected children would not exhibit severe symptoms.

Author Response

Point-by-Point response to reviewers

Respected Editor,

We have received revisions/suggestions for our submitted manuscript. All the concerns and suggestions of the reviewers have been addressed and we hope that the revised version of the manuscript will satisfy the concerns of all reviewers. We have attached a point-by-point response to each reviewer and a revised version of the manuscript for your consideration. Please let us know if any other changes are required in this regard. Last but not least, we are very thankful to the editor and all reviewers for their time and efforts in put their valuable suggestions. Indeed, their recommendations made this manuscript more scientifically elegant and sound.

Reviewer 2

General Query: Review of paper ‘Determination of Parent`s Intention to Vaccinate their Children 2 against COVID-19 and Barriers Associated with Childhood 3 COVID-19 Vaccination in Saudi Arabia – A Cross Sectional 4 Analysis’ by Y. H. Khan et al.

This study discussed child vaccination intake against COVID and the parental role in this regard; area of focus is Al-Jouf in Saudi Arabia. Among the sample size, 90% of parents were vaccinated, though only 42% of parents vaccinated their kids. Parental intention and potential barriers were mainly focused. It found a low intention among parents to vaccinate their kids against COVID-19 which depends on the education, profession and vaccination status of the parents. It is useful work and I recommend it after some revision.

Response: Respected reviewer, we are extremely grateful for your valuable comments and appreciation. We have tried our level best to address your comments and incorporate changes in manuscript. Kindly let us know if you need anything else. We hope that the revised version will satisfy all your concerns.

Main point:

Query 1: One issue here could be the sample size as only 444 parents took part in the survey.  Hence it is not a very robust result in terms of sample size. However, I believe results could be very similar if more samples are considered and the survey period is kept the same. Mention in the end that it is a limitation of this study.

Response 1: Respected reviewer, as per your recommendation, we have added limited sample size as a limitation of our study in the last paragraph of discussion section. Though we have achieved the estimated sample size, we agree with your comment that larger samples would provide more robust findings. Future studies should consider this limitation too.

Query 2: If there are comparison with one or two more locations of Saudi Arabia it could strengthen the findings. You cannot include that now but at least mention that part at the end also in the discussion section.

Response 2: Respected reviewer we have added comparisons with other studies conducted in different parts of KSA in the discussion section and highlighted these additions in yellow. We hope that the revised version will satisfy the concern.

Query 3: Mention the period of vaccination and the survey date of conducting data. This is because Saudi Arabia practically kept all vaccine doses very low since March 2022, compared to the previous period. (https://ourworldindata.org/explorers/coronavirus-data-explorer)

Response 3: Respected reviewer, the survey has been conducted during Feb-March 2022. This has been mentioned in the methodology section under sub-heading 2.2 (study design, settings, population). This was the time when Saudi government offered vaccination for those aged less than 12 years (mentioned in first paragraph of the discussion section).

Query 4: If you do similar analyses now and take the survey, most results will be much different.

Response 4: Respected reviewer, we have mentioned this point in the last paragraph of discussion section in which we are stating limitations of current study. We completely agree with you and hence this has already been mentioned that results should be considered with respect to the time at which study was conducted.

Query 5: Most of the countries worldwide are carefully monitoring vaccine success stories as claimed by the vaccine manufacturer. It is expected that all the observed data will be thoroughly investigated and analysed from time to time. As the data are available in the public domain it is easier for the authority to monitor the results easily. Many countries took decisions regarding vaccine doses and stopped vaccination. 

You may like to mention a few lines about why Saudi Arabia stopped the vaccine booster doses and see the figure below for cumulative doses. (https://ourworldindata.org/explorers/coronavirus-data-explorer).  Add few lines stating why many countries already stopped all vaccine doses including booster (Gibraltar, Iceland among others) or kept the dose nominal (Denmark, Israel etc.) since March 2022. See the trend for Israel and Denmark, which are practically nil since last March. This figure shows like many other countries, Saudi Arabia also stopped booster doses since March 2022. Discuss possible reasons why those countries took such stunts. Many of those countries were very proactive at the initial stages of vaccination.  

You discussed many possible causes of COVID vaccine hesitancy. Here also discuss possible causes of stopping all vaccine doses in many countries.

Response 5: Respected reviewer, thank you for highlighting this important issue. We completely agree with you that vaccination drive has not been vigorous now in many countries compared to how it started initially. We would like to point out two general points to elaborate our stance.

  1. Most of the developed countries started with rigorous initial vaccination of its citizens as soon as COVID-19 vaccination was available and most of its adult population fully vaccinated by now (71% KSA). As majority of the adult population has been vaccinated and children vaccination has been initiated (not compulsory), the number and severity of new COVID-19 cases has been drastically decreased (from thousands to only less than hundred now in KSA). High number of already vaccinated population along with less severity and number of new cases lead to low vaccination rate. However, COVID-19 vaccine is still available and given to anyone who wants it.
  2. The adult population in Saudi Arabia is required to have a Tawakalna application in their mobiles. This application provides vaccinated status proof. The color of the application will be green if a person has taken primary and booster doses of COVID-19 vaccines. A similar pattern of this application is followed for the children aged > 5 years. We did not come across any announcement from the ministry of health that pediatric vaccination against COVID-19 has been stopped for halted. We would like to inform that Saudi arabia has approved Moderna for kids having age 6 to 11 years at the end of April 2022 (source: https://www.saudigazette.com.sa/article/619779/SAUDI-ARABIA/Saudi-Arabia-approves-Moderna-vaccine-for-children-aged-6-11). In addition, a fourth dose was also recommended for people under 50 suffering from immunodeficiency (source: https://saudigazette.com.sa/article/620305/SAUDI-ARABIA/4th-COVID-19-dose-is-available-to-those-under-50-years-old-and-suffering-from-immunodeficiency).
  3. In a recent study conducted in Israel, authors found that repetitive booster doses are not beneficial against highly prevalent OMICRON variant. Authors found that level of antibodies required to protect against omicron variant are too high and cannot be achieved by vaccines.

https://www.reuters.com/world/middle-east/israeli-study-shows-4th-shot-covid-19-vaccine-not-able-block-omicron-2022-01-17/

  1. Vaccination has not been halted. Even in KSA and Israel, all these countries are vaccinating their children and the rate of vaccination in this group has so far been slow compared to adult vaccination that was mandatory.

https://www.timesofisrael.com/israel-gives-final-nod-to-covid-shots-for-under-5s-stops-short-of-recommendation/

  1. However, your concern is quite valid. We believe the lack of data reporting or limited reporting would be a reason that the graph of vaccination coverage is straight in publicly available data sources.

Query 6: You mentioned about Herd Immunity (line 225): “To achieve herd immunity and normalization of daily activities, at the end of June 2021, the Saudi government initiated compulsory vaccination of those aged between 12-18 years followed by massive vaccination of aged 5-12 years in February 2022.”In this regard Africa only had 47 % vaccination, much lower than herd immunity. Cases and deaths are very low. Moreover, Africa reduced vaccine doses drastically in the recent period and cases are showing a rapid decrease. (https://ourworldindata.org/explorers/coronavirus-data-explorer). On the other hand, many highly vaccinated countries those had a much high vaccination rates than the herd immunity threshold, performed much worse in the later period. Include some discussion about those important issues in the discussion.

Response 6: Respected reviewer, we agree that the vaccination rate in Africa in quite low as compared to many countries due to following reasons:

The response to multiple public health emergencies has affected COVID-19 vaccine rollout. Outbreaks of polio, measles, yellow fever and now Ebola have shifted priorities in the affected countries.

Difficult access to doses undermined vaccination efforts in 2021, however it has been improved now with countries on average receiving 67 doses per 100 people compared with 34 doses per 100 people at end 2021 and 13 doses per 100 at end September 2021.

At many countries with limited access to facilities, education and disease awareness, people no longer fear COVID-19 and so few are willing to get vaccinated. The number of people vaccinated has dropped significantly while the operational costs per person keeps increasing. This decline in effectiveness is due to sub-optimal planning and preparations, especially at the sub-national levels.

Despite all these hinderances, vaccination is Africa has not been stopped completely. To assist countries, intensify vaccination efforts, WHO in Africa has embarked on a raft of measures including supporting countries to assess the preparedness for vaccination campaigns at provincial and district levels, track vaccination among priority groups, carry out high-level advocacy to boost uptake, help countries integrate COVID-19 vaccines in other planned mass vaccination campaigns as well as deploy surge missions to countries to improve quality of vaccination drives.

We would like to clarify that the reporting of cases and deaths is widely compromised and criticized in African countries. We are recently collecting data on vaccination side effects from Sudan and Egypt and found that unfortunately 60% population is not vaccinated. This population has widespread beliefs on conspiracy theories and demonstrated untrust on health authorities.

Since achieving the herd immunity to combat the COVID-19 is well-defined strategy cum hypothesis, we are testing this hypothesis. However, it must be noted that COVID-19 pandemic control is not only relies on vaccination but also affect by various internal and external political, social and religious factors.

We hope that this explanation will satisfy the concern of the reviewer.

Query 7: Flu vaccine comparison in Table 1. The Influenza (Flu) vaccine was never compulsory for children in countries like UK and US. However, Europe and US are worst affected due to seasonal Flu every winter. I wonder why so many children in Saudi Arabia are vaccinated every year with the Flu vaccine. Kindly give brief statistics of Flu deaths in Saudi Arabia. Elaborate a few lines about Flu season, adult Flu vaccination rate, etc. 

Response 7: Respected reviewer, although KSA is a warm humid country but northern part of country becomes extremely cold during winters where temperature even drops to -12 degree Celsius in Turaif region with snow fall in many northern areas. There is an outbreak of upper respiratory tract infections at the start of winter, especially among young children. It is therefore recommended to have flu vaccination before arrival of winter and it is made available free of cost in all primary healthcare centers. According to WHO monthly influenza update (Sept 2022), Saudi Arabia ranked 3rd in Western Asian region with a total of 2404 enrolled cases (Please see the WHO report below - please refer to attachment as figure doesn't appear here). In order to control and monitor, WHO has also extended support to Saudi Arabia in testing upgraded electronic influenza surveillance platform that will help in timely collection, management and analysis of data to assist policy-makers in implementing efficient and timely preparedness and response measures.

https://www.emro.who.int/pandemic-epidemic-diseases/news/who-extends-support-to-saudi-arabia-in-testing-upgraded-electronic-influenza-surveillance-platform.html

The Ministry of health (MOH) Saudi Arabia recommends influenza vaccination.

https://www.moh.gov.sa/en/Ministry/MediaCenter/News/Pages/News-2020-10-24-004.aspx

According to the latest WHO data published in 2020 Influenza and Pneumonia Deaths in Saudi Arabia reached 6,132 or 4.58% of total deaths. The age adjusted Death Rate is 30.87 per 100,000 of population ranks Saudi Arabia #82 in the world

https://www.worldlifeexpectancy.com/saudi-arabia-influenza-pneumonia

A high flu vaccination rate in our study is related to extreme winter at the study location. The Al-Jouf regions is considered as one of the coldest and driest regions of the Saudi Arabia, making it vulnerable for flu. In this context, the ministry of health in Saudi Arabia arranges proactive campaigns for flu vaccines before winter where vaccination booths are arranged at parks, shopping malls, cross-ways and the public places.

Minor points:

Query 8: In line 245, Table 4: there is no Table 4 in the text.

Response 1: Respected reviewer, it was typing mistake, it has been corrected to table 1 and moreover, the number is also mentioned to avoid confusion.

Query 9: Figure 1 legend is not right. Rewrite it.

Response 9: Respected reviewer, we regret the wrong legend. We have changed it to “Methodological flowchart of study”.

Query 10: Line 268: rewrite this line “These results are in concordance with recently published literature where parents are concerned about adult vaccine side effects and are skeptical to use same the vaccines in their children”

Response 10: Respected reviewer, the line has been re-written as follows:

These findings are consistent with recently published studies where parents expressed concern about the side effects of adult immunizations and expressed skepticism about administering the same vaccines to their children.

Query 11: I like the point you mentioned about child vaccination in line 249: “However, the rate of parental acceptance of the COVID-19 vaccination was substantially lower (25%) among parents belonging to healthcare professions in our study. It might be because healthcare professionals are more informed than the general public regarding COVID-19 related low severity among children along with the fact that children are at minimal risk of mortality from COVID-19, and that the majority of infected children would not exhibit severe symptoms.”

Response 11: Respected reviewer, we are extremely grateful for your appreciation.

At the end, we all authors are very thankful for your time and efforts paid to review the current manuscript. In addition, we are also impressed with your skills, which provided us with opportunities to learn more. Indeed, your input contributed significant improvements in our draft. We welcome any other suggestions. Thank you.

Dr. Tauqeer Hussain Mallhi, PhD

Round 2

Reviewer 1 Report

The comments read as follows: 
  • Lines 2-3: Vaccination rather than vaccine.
  • Lines 44-45: The authors did not check the numbers provided here. The numbers are wrongly cited. Also, the reference link leads to SA data and not global data.
  • Regarding Response to Query 3: This test does not determine the association between two variables. It is suggested that authors use the appropriate test to determine association and to consult a statistician. This must be corrected in the entire manuscript, starting from the abstract throughout. This is very closely related to Authors' reply to Query 14.
  • Lines 77-85: The COVID-19 vaccine coverage rates are still not provided. Also, citing a newspaper website is not considered an official data source for this important matter of describing the situation regarding the COVID-19 vaccination in KSA.
  • Line 104: Even though authors responded that they did, they in fact did not clearly present both the response and participation rate.
  • Table 1: The authors said they checked and answered the comment regarding Table 1, but there were no changes made - the error remains. The sum of percentages is still not 100%, and this should be checked across all Tables.
  • The section Discussion still does not follow the logical order. This section always begins with a summary of the study's results, followed by paragraphs that compare the results with results of other similar studies and provide possible explanations for e.g. any observed differences.
  • Due to all of the previously stated reasons, as well as now authors stating in several instances that they addressed certain issues when in fact they did not, and particularly with the serious methodological issues and consequential wrong claims about associations, as explained above, the recommendation remains the same.

Author Response

Point-by-Point response to reviewers

Respected Editor,

We have received revisions/suggestions for our submitted manuscript. All the concerns and suggestions of the reviewers have been addressed and we hope that the revised version of the manuscript will satisfy the concerns of all reviewers. We have attached a point-by-point response to each reviewer and a revised version of the manuscript for your consideration. Please let us know if any other changes are required in this regard. Last but not least, we are very thankful to the editor and all reviewers for their time and efforts in put their valuable suggestions. Indeed, their recommendations made this manuscript more scientifically elegant and sound.

Reviewer 1

Query 1: Lines 2-3: Vaccination rather than vaccine.

Response 1: Respected Reviewer, we have made corrections in the title.

Query 2: Lines 44-45: The authors did not check the numbers provided here. The numbers are wrongly cited. Also, the reference link leads to SA data and not global data.

Response 2: Respected review, we have corrected reference no. 1. The corrected reference now represents global data. Following is also the screenshot of global estimates as of November 14, 2022 (left corner of the figure) [figure is given in attached response sheet as it is not displaying in online portal]. Our apology for the inconvenience. These estimates change every day, so it might be possible that you will find another close value on this link.

Query 3: Regarding Response to Query 3: This test does not determine the association between two variables. It is suggested that authors use the appropriate test to determine association and to consult a statistician. This must be corrected in the entire manuscript, starting from the abstract throughout. This is very closely related to the Authors' reply to Query 14.

Response 3: Respected reviewer, based on your recommendation, we have consulted a biostatistician who has advised us to consider some inferential tests in order to determine the factors associated with the childhood vaccination. We hope that the incorporation of new results will satisfy your concerns.

Query 4 Lines 77-85: The COVID-19 vaccine coverage rates are still not provided. Also, citing a newspaper website is not considered an official data source for this important matter of describing the situation regarding the COVID-19 vaccination in KSA.

Response 4: Respected reviewer, we have provided the Vaccination Coverage rate in the last paragraph of introduction section. The website of Saudi Gazette is an official website monitored and operated by the central ministries of Saudi Arabia. This website is meant to provide the information to Saudi Citizens and residents in English language. Since the official language for all communications in Saudi Arabia is in Arabic, the Saudi Gazette is considered most authentic source of English information for the public and other authorities. The ministry of health website does not provide recent statistics on the vaccination rate in the country and make most of its announcements either by a press conference or through official news agency (Saudi Gazette). However, there is another website that primarily focuses on the number of COVID-19 cases in the country along with information on the number of primary and booster doses administered (https://covid19.moh.gov.sa/). We have provided the reference of this website in the last paragraph of the introduction section.

However, we have furnished the important references with the ministry of health links. We hope that the addition of new authentic sources will satisfy your concerns.

Query 5: Line 104: Even though authors responded that they did, they in fact did not clearly present both the response and participation rate.

Response 5: Respected reviewer, we have now provided details on the response and participation rate in the results section. Please let us know if any other information is needed in this regard.

Query 6: Table 1: The authors said they checked and answered the comment regarding Table 1, but there were no changes made - the error remains. The sum of percentages is still not 100%, and this should be checked across all Tables.

Response 6: Respected Review, please accept our apology for error that appeared during rounding off the decimals. We have corrected all the tables and double checked by two authors to ensure accuracy.

Query 7: The section Discussion still does not follow the logical order. This section always begins with a summary of the study's results, followed by paragraphs that compare the results with results of other similar studies and provide possible explanations for e.g. any observed differences.

Response 7: Respected reviewer, based on your suggestion we have re-written the discussion section and we hope that the revised version of the discussion will satisfy your concerns. Please let us know if any further update is needed.

Query 8: Due to all of the previously stated reasons, as well as now authors stating in several instances that they addressed certain issues when in fact they did not, and particularly with the serious methodological issues and consequential wrong claims about associations, as explained above, the recommendation remains the same.

Response 8: Respected reviewer, we have tried our best to address all the suggestions and concerns related to our submission. However, we did our best to address all your concerns during the last round of revisions, but it might be possible that we misunderstood some of the queries. We hope that the revised version of the manuscript will satisfy all the queries. Since we believe that a great subject expert like you provides opportunities to learn more new things, we welcome any other correction or suggestion to improve our manuscript.

At the end, we all authors are very thankful for your time and efforts paid to review the current manuscript. In addition, we are also impressed with your skills related to statistics and citations which provided us with a window to learn new things. Indeed, your input contributed significant improvements in our draft. We welcome any other suggestions. Thank you.

Dr. Tauqeer Hussain Mallhi, PhD

Round 3

Reviewer 1 Report

  • Regarding Query 2 - In their Response 2, the authors state that the Reviewer should acknowledge that the number of vaccinated people changes every day. What the Authors do not state is the following the - first two versions of their manuscript had an error, in both versions the Authors provided wrong numbers for the number of fully vaccinated people and people vaccinated with at least one dose, which can easily be checked. In the first two versions of their manuscript authors wrongly presented these numbers, as if the number of fully vaccinated was higher than those receiving at least one dose - whatever the date you choose to look at. Further on - the Authors were obviously aware of e what my comment referred to - since finally in this third version they revised and corrected this, but without acknowledging their error in the response, or marking it with yellow in their manuscript as they have done with other corrections.
  • The Authors have made a mess with the statistical methodology in this manuscript in the first version of the paper they stated that they have done a multivariate analysis but they did not provide the results, in the second version of the manuscript the Authors removed the sentence about multivariate analysis from the section methods. Finally, in this last third version of the paper, the Authors added one sentence in the section Methods where they mention binary logistic regression. Finally, in this last third version of the paper, the Authors add a completely new Table 4 which presents the results of the univariate and multivariate analysis. The new Table 4 is very important because it significantly improves the quality of this paper. The remark relates to the need to align the sections Methods and Results in a way that in the subsection for statistical analysis in the Methods section you state that multivariate analysis was done too. Explain at what p level of significance from the univariate analysis did you consider variables for inclusion in the multivariate analysis. Also, since on the new Table 4 there is a mention of adjusted OR, it is necessary to add to the subsection for statistical analysis a description for which variables was the adjustment done and according to what criteria were those variables chosen for the adjustment.
  • Since now you have the results of the multivariate regression analysis, they should be mentioned in the Abstract.
  • The English language of the manuscript needs to be improved in the area of newly added text.
  • I would like to thank the Authors for their extreme effort that they put in to improve their manuscript, and in particular for adding the new Table 4 with very useful information which has significantly improved the quality of their manuscript.

Author Response

Query 1: Regarding Query 2 - In their Response 2, the authors state that the Reviewer should acknowledge that the number of vaccinated people changes every day. What the Authors do not state is the following the - first two versions of their manuscript had an error, in both versions the Authors provided wrong numbers for the number of fully vaccinated people and people vaccinated with at least one dose, which can easily be checked. In the first two versions of their manuscript authors wrongly presented these numbers, as if the number of fully vaccinated was higher than those receiving at least one dose - whatever the date you choose to look at. Further on - the Authors were obviously aware of e what my comment referred to - since finally in this third version they revised and corrected this, but without acknowledging their error in the response, or marking it with yellow in their manuscript as they have done with other corrections.

Response 1: Respected Reviewer, thank you very much for your comment and providing us an opportunity to express our views on this comment. First of all, we pay extreme apology if we misunderstood any of your comments during previous rounds of revisions. Secondly, please accept our apology for stating wrong numbers in previous versions. It was overlooked even after your indication towards this error. We agree that we replace numbers between fully vaccinated and those vaccinated with one dose. We assure this it was an honest error and please accept our apology for this inconvenience. We all authors acknowledge this error.

Query 2: The Authors have made a mess with the statistical methodology in this manuscript in the first version of the paper they stated that they have done a multivariate analysis but they did not provide the results, in the second version of the manuscript the Authors removed the sentence about multivariate analysis from the section methods. Finally, in this last third version of the paper, the Authors added one sentence in the section Methods where they mention binary logistic regression. Finally, in this last third version of the paper, the Authors add a completely new Table 4 which presents the results of the univariate and multivariate analysis. The new Table 4 is very important because it significantly improves the quality of this paper. The remark relates to the need to align the sections Methods and Results in a way that in the subsection for statistical analysis in the Methods section you state that multivariate analysis was done too. Explain at what p level of significance from the univariate analysis did you consider variables for inclusion in the multivariate analysis. Also, since on the new Table 4 there is a mention of adjusted OR, it is necessary to add to the subsection for statistical analysis a description for which variables was the adjustment done and according to what criteria were those variables chosen for the adjustment. 

Response 2: Respected review, we agree that previous version of the manuscript lack logistic regression even it was stated in the methods section. However, your suggestions provided us opportunity to correct our mistake. We carried out logistic regression analysis and presented table with useful information. Thank you very much for appreciating this addition. In fact, we all authors are very thankful to you because it’s your valuable suggestion that resulted in improved quality of the manuscript.

We have revised section 2.6 (statistical analysis) in light of your comments. It now contains detailed information about univariate as well as multivariate analysis. Furthermore, we have also explained how the covariates were selected to be entered in multivariate regression analysis. Regarding the adjusted odds ratio (AOR) presented in Table 4, AOR holds other relevant variables constant and provides the odds ratio for the potential variable of interest which is adjusted for the other independent variables in the logistic model. We did not use stratification technique in the analysis. We have revised the entire section 3.3 to make it clearer and more comprehensible for the readers. We hope this is appropriate now.

“As shown in Table 4, nine independent variables were found to be significantly associated with our dependent variable in the univariate analysis (parents’ education level, profession, COVID-19 vaccination status, their involvement in healthcare decision of children, efficacy and safety concerns, family’s pressure to avoid COVID-19 vaccine, belief children were not at risk of the infection (low perceived vulnerability), and belief that conventional prevention measures were enough to protect children from the infection). Consequently, these covariates were simultaneously entered in the model. Results indicated that our model was statistically significant (χ2 (11) = 74.744, p < 0.001) and had good predictive capability (Hosmer-Lameshow test: χ2 (8) = 7.051, p = 0.531). Of all the predictors in the model, only four were found to be significant predictors of parental intention to vaccinate children for COVID-19 (Table 4). Increased parent’s education level was associated with positive intention to vaccinate children against the disease, keeping all other covariates constant (secondary education: OR = 3.617, p = 0.010; tertiary education: OR = 2.775, p = 0.042). Furthermore, parents who were vaccinated for COVID-19 were 7-times more likely to get their children vaccinated for COVID-19, holding other independent variables constant. Additionally, mother’s involvement in child’s healthcare-related decision making (OR 4.353, p < 0.001) as well as father’s and mother mutual decision making (OR 3.195, p < 0.001) were associated with a higher likelihood of parental COVID-19 vaccine intention. Last but not the least, parent’s trust on COVID-19 vaccine’s safety was a significant factor for COVID-19 pediatric vaccination (OR 2.483, p = 0.022).”

Query 3: Since now you have the results of the multivariate regression analysis, they should be mentioned in the Abstract. 

Response 3: Respected reviewer, multivariate results have been incorporated in the abstract as per your comments.

Query 4 The English language of the manuscript needs to be improved in the area of newly added text.

Response 4: Respected reviewer, we have improved the English of newly added text in the manuscript. We hope this is acceptable now.

Query 5: I would like to thank the Authors for their extreme effort that they put in to improve their manuscript, and in particular for adding the new Table 4 with very useful information which has significantly improved the quality of their manuscript. 

Response 5: Respected reviewer, in fact we all authors are greatly thankful to you. It was a productive and resourceful interaction with you. We have experienced a true essence of peer-review. Thank you very much for valuables comments that improve our manuscript.